# Photosynthesis-inspired H$_2$ generation using a chlorophyll-loaded liposomal nanoplatform to detect and scavenge excess ROS

Wei-Lin Wan[1,4], Bo Tian[1,4], Yu-Jung Lin[1], Chiranjeevi Korupalli[1], Ming-Yen Lu[2], Qinghua Cui[1], Dehui Wan[1], Yen Chang[3]* & Hsing-Wen Sung [1]*

A disturbance of reactive oxygen species (ROS) homeostasis may cause the pathogenesis of many diseases. Inspired by natural photosynthesis, this work proposes a photo-driven H$_2$-evolving liposomal nanoplatform (Lip NP) that comprises an upconversion nanoparticle (UCNP) that is conjugated with gold nanoparticles (AuNPs) via a ROS-responsive linker, which is encapsulated inside the liposomal system in which the lipid bilayer embeds chlorophyll $a$ (Chl$a$). The UCNP functions as a transducer, converting NIR light into upconversion luminescence for simultaneous imaging and therapy in situ. Functioning as light-harvesting antennas, AuNPs are used to detect the local concentration of ROS for FRET biosensing, while the Chl$a$ activates the photosynthesis of H$_2$ gas to scavenge local excess ROS. The results thus obtained indicate the potential of using the Lip NPs in the analysis of biological tissues, restoring their ROS homeostasis, possibly preventing the initiation and progression of diseases.

---

[1] Department of Chemical Engineering and Institute of Biomedical Engineering, Frontier Research Center on Fundamental and Applied Sciences of Matters, National Tsing Hua University, Hsinchu, Taiwan, ROC. [2] Department of Materials Science and Engineering, National Tsing Hua University, Hsinchu, Taiwan, ROC. [3] Taipei Tzu Chi Hospital, Buddhist Tzu Chi Medical Foundation and School of Medicine, Tzu Chi University, Hualien, Taiwan, ROC. [4]These authors contributed equally: Wei-Lin Wan, Bo Tian. *email: ychang@tzuchi.com.tw; hwsung@mx.nthu.edu.tw

Many pathogenic processes are involved in the overproduction of reactive oxygen species (ROS), including hydroxyl radical ($\bullet$OH), peroxynitrite ($ONOO^-$), and hydrogen peroxide ($H_2O_2$). Physiologically, the ROS that are produced intracellularly regulate cell signaling, modulate protein functions, and mediate inflammation[1,2]. However, the excess ROS that are generated in inflammatory cells, such as macrophages, can damage cellular proteins, DNA, and lipids. Such an imbalance in ROS production may cause the pathogenesis of numerous human diseases, including cancer, cardiovascular disorder, and diabetes[3,4].

Hydrogen ($H_2$), which has potential as an antioxidant, is known to be able selectively to reduce concentrations of highly cytotoxic ROS, including $\bullet$OH and $ONOO^-$, in diseased cells while preserving the physiological functions of other ROS in normal cells[5,6]. Moreover, since it is smaller than those of other antioxidants, $H_2$ can readily diffuse into cells and tissues where it performs its therapeutic functions[6]. Owing to its unique ability to regulate ROS homeostasis, $H_2$-gas therapy has recently received considerable attention[7–9]. However, the concentration of $H_2$ that can be delivered to diseased tissues by the traditional administration routes is typically well below its therapeutic threshold for scavenging locally generated excess ROS, owing to its low solubility in body fluids[10].

Inspired by natural photosynthesis, this work proposes a nanocomplex that consists of a lanthanide-doped upconversion nanoparticle (UCNP; $NaYbF_4:Er@CaF_2$) that is conjugated with gold nanoparticles (AuNPs) via a ROS-responsive thioketal (TK)-based linker, which is encapsulated in a liposomal system in which the lipid bilayer embeds with chlorophyll a (Chla). UCNPs have been used as an excellent donor fluorophore in Förster resonance energy transfer (FRET) for biological detection[11–13], and AuNPs have been used as an acceptor fluorophore in FRET biosensing[14,15].

Figure 1 depicts the composition/structure of an as-proposed Lip NP and the mechanisms by which it concurrently detects and scavenges overproduced ROS in situ, modulating ROS homeostasis. To make the UCNP hydrophilic for use in water, its surface is modified with citrate (Cit-UCNP), which is also an electron donor, providing electrons as well as protons[16]. The Cit-UCNP can potentially serve as a remotely controlled transducer, converting tissue-penetrating near infrared (NIR) radiation (980 nm) into upconversion luminescence (UCL) at around 550 nm (green UCL) and 660 nm (red UCL). In contrast, the AuNPs that are conjugated with the Cit-UCNP are used as light-harvesting antennas that detect the local ROS concentration.

In a physiological state, the TK bond of the nanocomplex (Cit-UCNP-TK-AuNPs) is intact, and the distance between the Cit-UCNP (donor fluorophore) and the AuNPs (acceptor fluorophore) in the nanocomplex is sufficiently short to allow FRET (FRET on). Hence, upon excitation by 980 nm NIR light, the AuNPs absorb the green UCL emission of the Cit-UCNP.

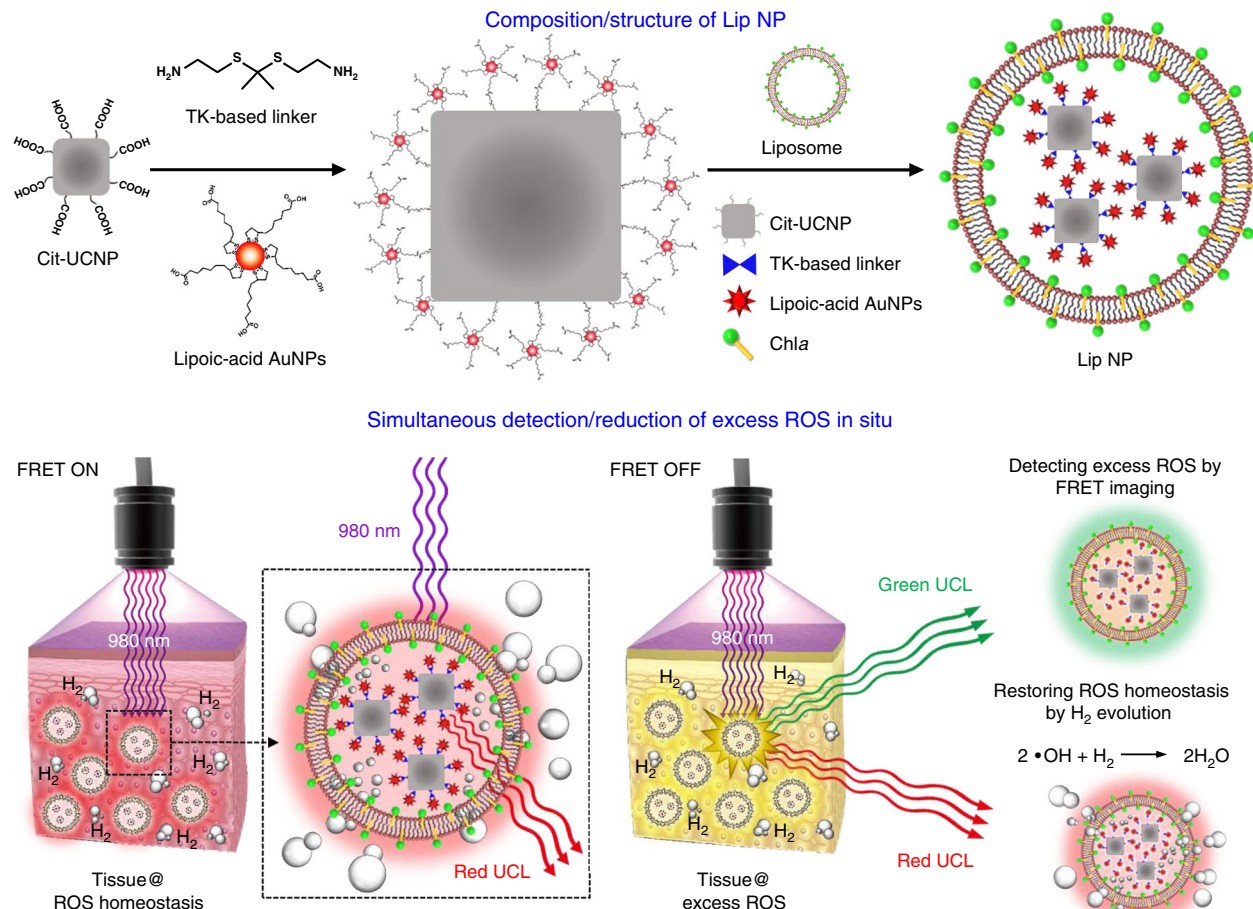

**Fig. 1 Composition/structure of Lip NP and its operating mechanism.** In as-proposed Lip NP, whose aqueous core encapsulates Cit-UCNP-TK-AuNPs nanocomplexes and in which the lipid bilayer embeds chlorophyll a (Chla). NIR laser penetrates biological tissue and is converted to green and red UCL by Cit-UCNP in nanocomplex. Green UCL is used to measure local ROS concentration for FRET imaging, and red UCL induces photosynthesis of gaseous $H_2$ to scavenge excess ROS.

However, under oxidative stress conditions, the ROS homeostasis is altered and the excess ROS cleaves the TK bond of the nano-complex, dissociating the Cit-UCNP from the AuNPs (FRET off), ultimately yielding green UCL at 550 nm. Therefore, this FRET imaging technique can be powerful for detecting excess ROS in biological tissues.

Figure 2 presents a potential mechanism of the photocatalytic evolution of hydrogen. When red UCL at 660 nm is harvested by Chl*a* (a photosensitizer), the latter becomes excited (Chl*a**)[17–19]. The photo-excited electrons that are released from the Chl*a** are rapidly transferred to the AuNPs (an electron acceptor and a proton-reducing catalyst) that are conjugated with the Cit-UCNP. The AuNPs then collect protons from citrate (a sacrificial electron donor), which caps the UCNP, and hydrogen is thus evolved, locally scavenging the excess ROS[20,21]. The oxidized Chl*a* (after the loss of an electron, Chl*a*+) can be reduced by its acceptance of an electron from citrate, returning to its ground state[22]. Photocatalytic hydrogen production, which typically involves a photosensitizer, a proton-reducing catalyst, and a sacrificial electron donor, has been widely used in artificial photosynthetic systems for the efficient utilization of solar energy, solving energy-related and environmental problems[23–25].

## Results

**Characteristics of nanocomplexes and Lip NPs.** The oleic acid-capped UCNPs (OA-UCNPs), which were hydrophobic and could form a colloidal solution in cyclohexane (Fig. 3a), were synthesized by a thermolytic method[26]. Following treatment with citrate (Cit-UCNPs) using a ligand exchange method[27], the monodispersed nanocubes, which had a mean size of ca. 20 nm, became hydrophilic and dispersed effectively in water. Upon excitation with a 980 nm NIR laser, a strong UCL, appearing yellow-red because it combined green and red emissions, from a colloidal aqueous solution of Cit-UCNPs, was clearly visible (Fig. 3a).

According to the Fourier-transform infrared (FT-IR) spectra (Fig. 3b), the sample of OA-UCNPs yielded two characteristic peaks at 1559 and 1453 cm$^{-1}$, representing the asymmetric and symmetric stretching vibrations of the carboxylate ions in the capping OA, respectively. However, these peaks were shifted to 1589 and 1401 cm$^{-1}$, respectively, for the sample of Cit-UCNPs, revealing that the OA ligands on the surface of UCNPs were

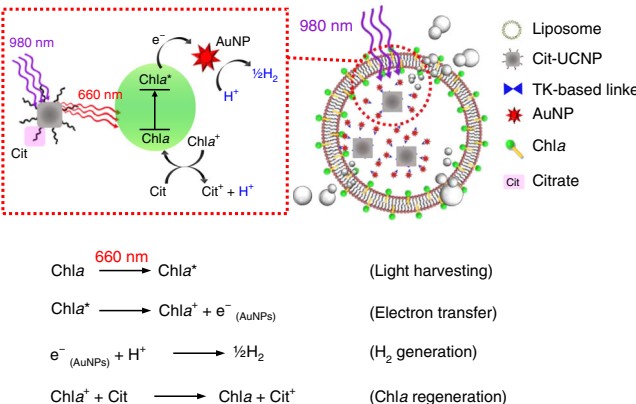

| | |
|---|---|
| Chl*a* $\xrightarrow{660 \text{ nm}}$ Chl*a** | (Light harvesting) |
| Chl*a** $\longrightarrow$ Chl*a*+ + e$^-$ (AuNPs) | (Electron transfer) |
| e$^-$ (AuNPs) + H$^+$ $\longrightarrow$ ½H$_2$ | (H$_2$ generation) |
| Chl*a*+ + Cit $\longrightarrow$ Chl*a* + Cit$^+$ | (Chl*a* regeneration) |

**Fig. 2 Potential mechanism of photosynthesis of hydrogen.** When red UCL at 660 nm is harvested by Chl*a*, the latter becomes excited (Chl*a**). The photo-excited electrons are rapidly transferred to AuNPs, which can collect protons from citrate, resulting in hydrogen evolution. The oxidized Chl*a* can be reduced with acceptance of an electron from citrate, returning to its ground state.

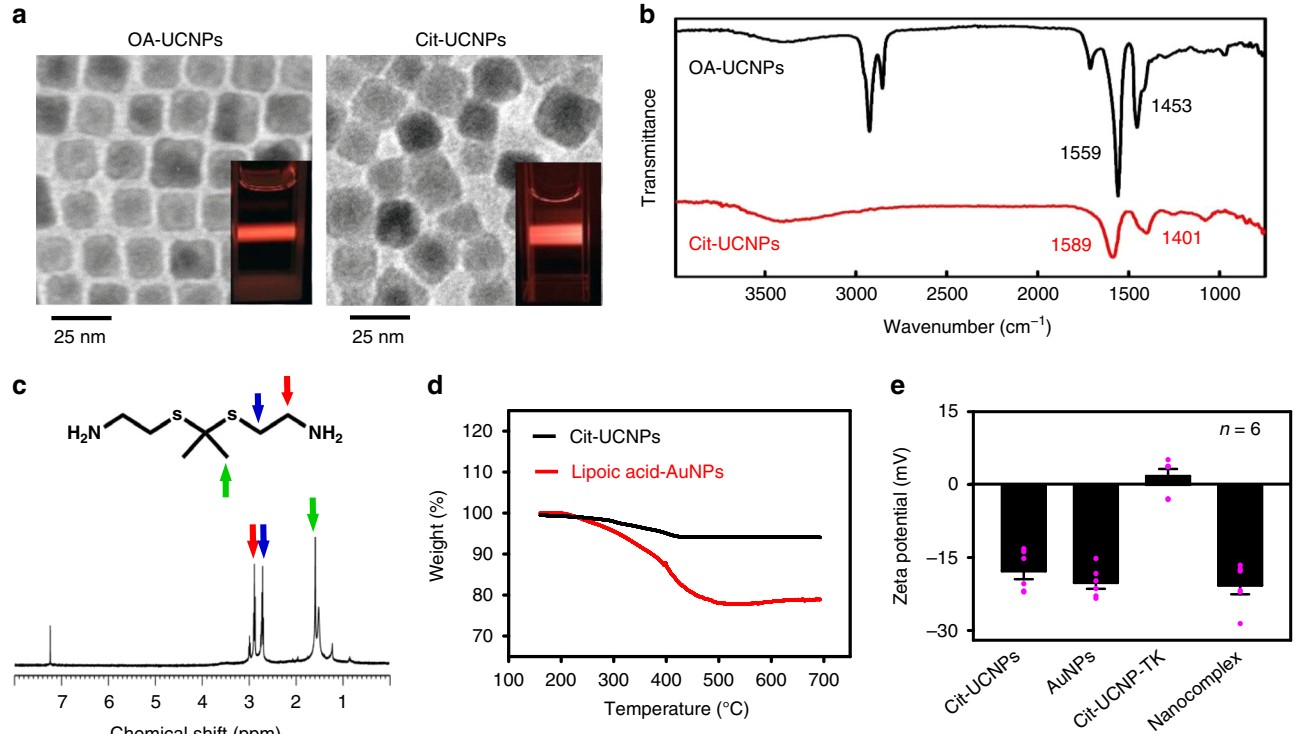

**Fig. 3 Characteristics of UCNPs, TK-based linker and nanocomplexes. a** TEM images of OA-UCNPs and Cit-UCNPs, and their corresponding emission images under NIR laser irradiation. **b** FT-IR spectra of OA-UCNPs and Cit-UCNPs. **c** $^1$H NMR spectrum of TK-based linker. **d** TGA thermograms of Cit-UCNPs and lipoic acid-capped AuNPs. **e** Zeta potentials of Cit-UCNPs, AuNPs, Cit-UCNP-TK, and nanocomplexes. Data in (**e**) are represented as mean ± SE. Each pink dot represents one observed data point. Source data are provided as Source Data file.

replaced by the Cit ligands. The ROS-responsive TK-containing linker was synthesized using a procedure that can be found elsewhere[28], which was verified by [1]H NMR spectroscopy. The characteristic peaks at ~1.58, 2.74, and 2.98 ppm corresponded to the protons in –CH$_3$, –CH$_2$–S, and –CH$_2$–N, respectively, in the TK-containing linker (Fig. 3c). The AuNPs used herein, which were capped with lipoic acid and had a diameter of ca. 5.5 nm, were obtained commercially. The results of thermogravimetric analysis (TGA) showed that the amount of the lipoic acid (Cit) ligands that was functionalized on the surface of AuNPs (UCNPs) was 22.2 (6.0) wt% (Fig. 3d).

The nanocomplexes were prepared by a standard coupling reaction in which the carboxyl groups from the Cit-UCNP or AuNPs were conjugated with the amine groups from the TK-based linker in the presence of EDC/NHS. Zeta potential measurements indicate that the Cit-UCNPs were negatively charged (Fig. 3e), and the zeta potential varied from –17.8 to 1.6 mV after they were coupled with the TK-based linker (Cit-UCNP-TK); upon AuNP (−20.2 mV) conjugation, the zeta potential was positively shifted to −21.0 mV, suggesting the successful preparation of nanocomplexes.

The morphologies of the as-prepared nanocomplexes in the absence/presence of ROS (50 μM H$_2$O$_2$) were studied by scanning transmission electron microscopy (STEM). ROS in solution is known to be reactive and so has a short half-life[29]. In cells, enzymatic and nonenzymatic reactions can convert ROS to H$_2$O$_2$, which has a relatively long half-life and can diffuse out of the cells, making H$_2$O$_2$ a good marker of oxidative stress[30,31]. Local extracellular concentrations of H$_2$O$_2$ under normal physiological conditions are in the range of 0.5–7 μM, while those under physiological conditions are elevated as high as 10–50 μM[32,33].

According to Fig. 4a, in the absence of ROS, the structure of the conjugated AuNPs on UCNP, which had a mean size of ca. 30 nm, was clearly seen in the STEM image, while AuNPs were dissociated from UCNP in the presence of ROS. The energy-dispersive X-ray (EDX) spectroscopic linescan that was

conducted using STEM on a nanocomplex sample in the absence of ROS revealed a higher Au concentration in the peripheral region (AuNPs) and higher concentrations of Yb, F, and Ca in the central region (UCNP); these findings are highly consistent with the designed structure of the conjugate nanocomplex. Conversely, in the presence of ROS, the signals of AuNPs in the elemental linescan disappeared, suggesting that the TK-containing bond of the nanocomplex was cleaved.

H$_2$-generating Lip NPs were prepared using a thin-film hydration technique in the presence of nanocomplexes and Chl$a$. As verified by STEM, nanocomplexes were successfully encapsulated in the aqueous core of the Lip NPs (Fig. 4b, indicated by yellow arrowheads), which had a size distribution of 150–250 nm and a zeta potential of ca. 0.2 mV. Confocal laser scanning microscopy (CLSM) revealed that Chl$a$ (green color) was embedded in their lipid membranes (Fig. 4c). The Lip NP formulation was optimized by maximizing the content of each component that could be encapsulated or the amount of H$_2$ gas that could be generated. The encapsulated concentrations of AuNPs in the nanocomplexes and Chl$a$ in the as-optimized Lip NPs were determined to be 29.1 ± 1.0 nM and 38.4 ± 8.5 μM, respectively (mean ± SE, $n = 6$ batches).

**H$_2$O$_2$ detection and H$_2$ generation**. The optical properties of plain Cit-UCNPs upon NIR light excitation and plain AuNPs were investigated by fluorescence spectrophotometry and UV/vis spectrophotometry, respectively. According to Fig. 5a, the large spectral overlap between the emission wavelength of Cit-UCNPs and the absorbance wavelength of AuNPs suggests that the AuNPs in the nanocomplexes can be activated by their conjugated Cit-UCNP-emitted green UCL, probably allowing efficient FRET.

To determine whether the Cit-UCNP (as the transducer) and AuNPs (as light-harvesting antennas) in the conjugate nanocomplexes can form an efficient FRET imaging system for

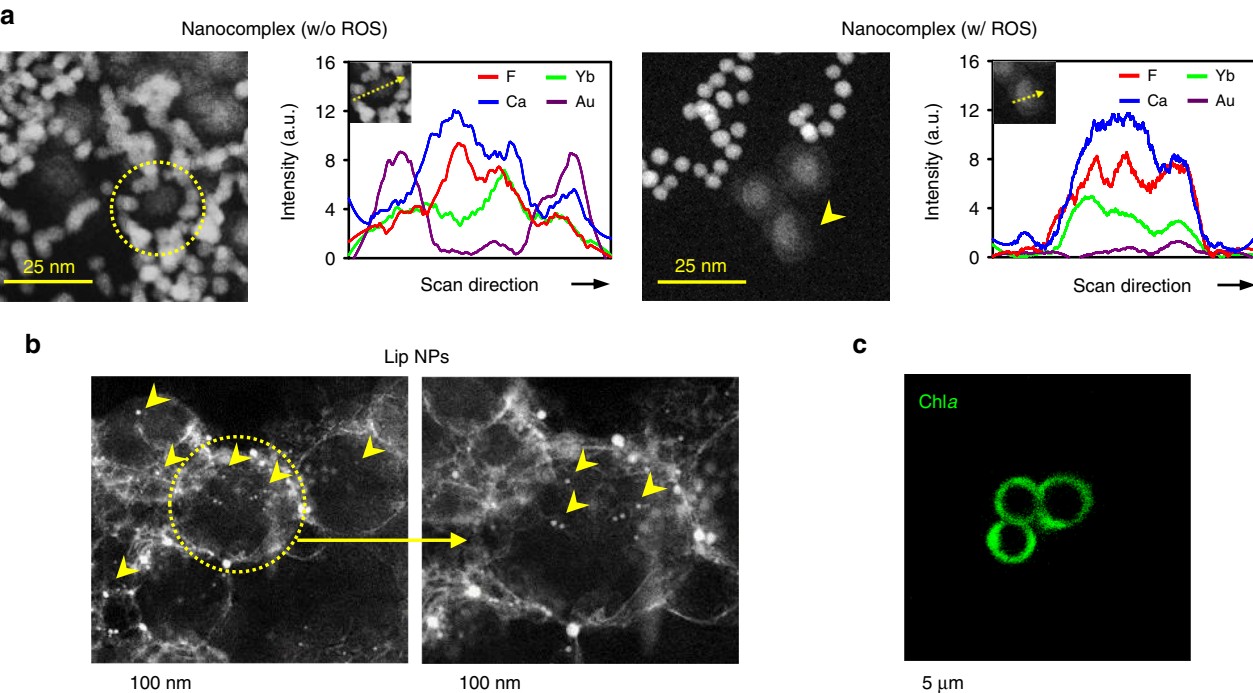

**Fig. 4 ROS-sensitivity of nanocomplexes and structures of Lip NPs. a** STEM images and elemental linescans of nanocomplexes in the absence/presence of ROS. **b** STEM images and **c** CLSM image of Lip NPs. Owing to the limited optical resolution in CLSM, Lip NPs that had undergone centrifugation and had diameters (2–3 μm) were observed. Source data are provided as Source Data file.

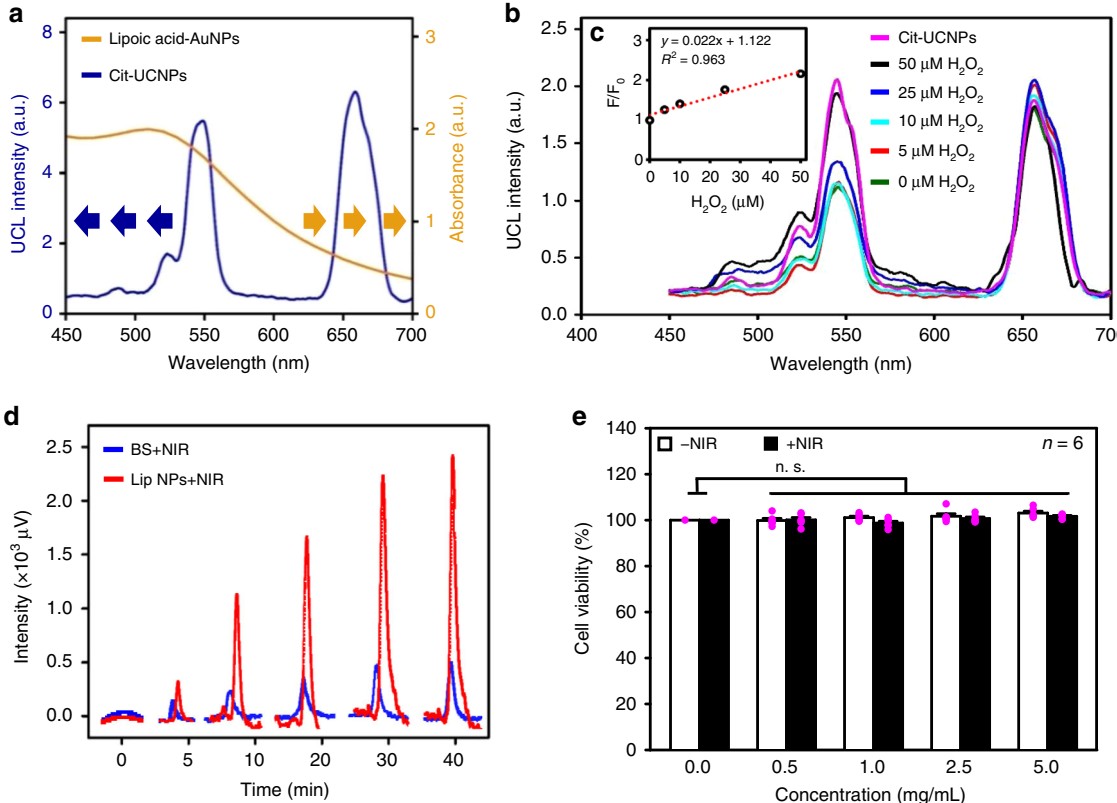

**Fig. 5 H₂O₂ detection, H₂ generation, and cytotoxicty. a** Fluorescence spectrum of Cit-UCNPs irradiated at 980 nm and UV/vis absorbance of AuNPs. **b** Fluorescence excitation spectra of nanocomplexes following incubation with various concentrations of H₂O₂ for 30 min. **c** Linear correlation curve of green UCL ($F/F_0$) intensity against concentration of H₂O₂. **d** Cumulative amount of gaseous H₂ generated from BS or Lip NPs under NIR laser irradiation. **e** Cell viability of RAW264.7 cells incubated with various concentrations of Lip NPs. Data in (**e**) are represented as mean ± SE. *P* values were determined using the two-tailed unpaired Student's *t* test. n.s. not significant. Each pink dot represents one observed data point. Source data are provided as Source Data file.

detecting H₂O₂, the fluorescence spectra of the nanocomplexes that were generated in response to various concentrations of H₂O₂, upon NIR laser irradiation, were recorded. As shown in Fig. 5b, the emission intensity of green UCL recovered more as the H₂O₂ concentration increased, whereas that of the red UCL at 660 nm remained relatively unchanged. These analytical results suggest that the nanocomplexes respond to low H₂O₂ and so may be an effective ultra-sensitive ROS-responsive FRET imaging system.

In this study, a correlation curve is obtained by plotting the relative intensity of the green UCL ($F/F_0$) of the nanocomplexes as a function of H₂O₂ concentration, where $F_0$ and $F$ represent the intensities of green UCL in the absence and presence of H₂O₂, respectively. According to Fig. 5c, the relative green UCL intensity varied linearly with the local H₂O₂ concentration (10–50 μM), which is in the detection range of disease-relevant H₂O₂ levels[32,33], suggesting that the nanocomplexes may be applied to detect trace amounts of H₂O₂ that are produced under pathological conditions.

Motivated by the aforementioned results, we evaluated the capacity of the Lip NPs to produce H₂ gas by NIR-to-vis-driven photosynthesis. A bulk solution (BS) that contained the free reacting molecules, nanocomplexes and Chl*a*, at equivalent concentrations was used as a control. The profiles of the gaseous H₂ that accumulated during NIR laser exposure were obtained using gas chromatography. According to Fig. 5d, upon exposure to the NIR laser (+NIR), the concentrations of H₂ that accumulated from the Lip NPs consistently exceeded those from BS. These findings demonstrate that the nanotransducer (the Cit-UCNP in the conjugate nanocomplexes) in the Lip NPs

absorbed NIR light and transferred the excitation energy to their light-harvesting Chl*a* more effectively than the free reacting molecules that were suspended in BS, ultimately yielding more gaseous H₂.

The sizes of Lip NPs before and after they had been treated with NIR laser irradiation and/or H₂O₂ did not differ significantly ($P > 0.05$, Supplementary Table 1), suggesting that the as-developed liposomal formulation was stable following such treatments. Results of the cytotoxicity study reveal that the Lip NPs, without (−NIR) or with NIR laser (+NIR) excitation, had no toxicity (Fig. 5e), and so can be used to scavenge the excess ROS in cells.

**In vitro ROS scavenging and anti-inflammation.** The feasibility of using the Lip NPs to scavenge the overproduced ROS in macrophages (RAW264.7) that had been stimulated by lipopolysaccharide (LPS) was assessed. LPS is known to be a potent activator of macrophages, promoting the production of ROS, including •OH, ONOO⁻, and H₂O₂, as well as the expressions of many proinflammatory cytokines, such as interleukin (IL)-1β and IL-6[34,35]. The BS that contained equal concentrations of free nanocomplexes and Chl*a* served as a control. To determine the cellular ROS levels, cells were stained with CellROX Deep Red. Double immunohistochemistry staining was carried out to visualize the intracellular expressions of proinflammatory cytokines IL-1β and IL-6. The amount of intracellular ROS was measured using the 2′,7′-dichlorofluorescin diacetate (DCFDA) assay kit, and the expression levels of proinflammatory cytokines in the cells were determined by ELISA[8].

As shown in Fig. 6a, Lip NPs + NIR effectively reduced the excess production of ROS in macrophages in a concentration-dependent manner ($P < 0.05$, unpaired Student's $t$ test). The highest Lip NP concentration that was used in this experiment was 5.0 mg/mL, but the reduction of ROS overproduction was maximized at a Lip NP concentration of 1.0 mg/mL, which was therefore used in subsequent studies.

According to Fig. 6b–d, the LPS-stimulated macrophages noticeably overproduced ROS, relative to the unstimulated control cells, triggering excess expressions of proinflammatory cytokines IL-1β and IL-6 ($P < 0.05$, unpaired Student's $t$ test). Treatment with BS + NIR or Lip NPs + NIR significantly suppressed the over-productions of ROS and proinflammatory cytokines ($P < 0.05$, unpaired Student's $t$ test). Notably, the overproductions of the LPS-stimulated ROS and proinflammatory cytokines in the group that was treated with Lip NPs + NIR were reduced by more than those in the group that was treated with BS + NIR, likely because the amount of $H_2$ produced in the former case significantly exceeded that in the latter case (Fig. 5d).

The levels of $H_2O_2$ that remained in the test media following various treatments were estimated from the linear correlation between the relative green UCL intensity and the local $H_2O_2$ concentration, which was obtained using the FRET imaging technique, as described above (Fig. 5c). According to Fig. 6e, treatment with BS + NIR led to a 28.2 ± 4.4% reduction in $H_2O_2$ level relative to that of the LPS-stimulated cells, while treatment with Lip NPs + NIR caused a 65.7 ± 3.7% reduction in $H_2O_2$ level. To verify the $H_2O_2$ concentration in cell culture systems,

an additional experiment using a commercially available assay kit was performed to measure the actual extracellular $H_2O_2$ concentrations. According to Supplementary Fig. 1, the trends that were obtained using these two different methods were similar, suggesting that the FRET-based correlation method that was proposed in the study can be used to estimate ROS levels in cell culture systems.

**Ex vivo ROS detection and $H_2$ generation.** Finally, a light penetration experiment was performed using porcine skin tissues that had been injected with Cit-UCNPs (Fig. 7a) or Lip NPs (Fig. 7b). Test samples under irradiation with an NIR laser (980 nm) were studied with a fluorescence microscope. Limited by the gap between the objective lens of the fluorescence microscope and its sample stage, the thickness of the porcine tissues used in this investigation was approximately 2 mm. According to Fig. 7a, while the background autofluorescence level was negligible, the NIR laser completely penetrated the tissue sample and was effectively converted to green UCL by the NIR-to-vis-excited Cit-UCNPs. The tissue penetration depth of a green laser is reported only approximately 0.3 mm[36].

From the NIR-irradiated tissue samples that had been injected with the Lip NPs, no significant fluorescence signal was observed in the absence of $H_2O_2$ (FRET on, Fig. 7b), whereas high-contrast green UCL was easily detected in the presence of 50 μM $H_2O_2$ (FRET off), indicating the great potential of Lip NPs sensitively to detect $H_2O_2$ in biological tissues. The test tissue samples in the absence/presence of $H_2O_2$ were, however, found to evolve gaseous

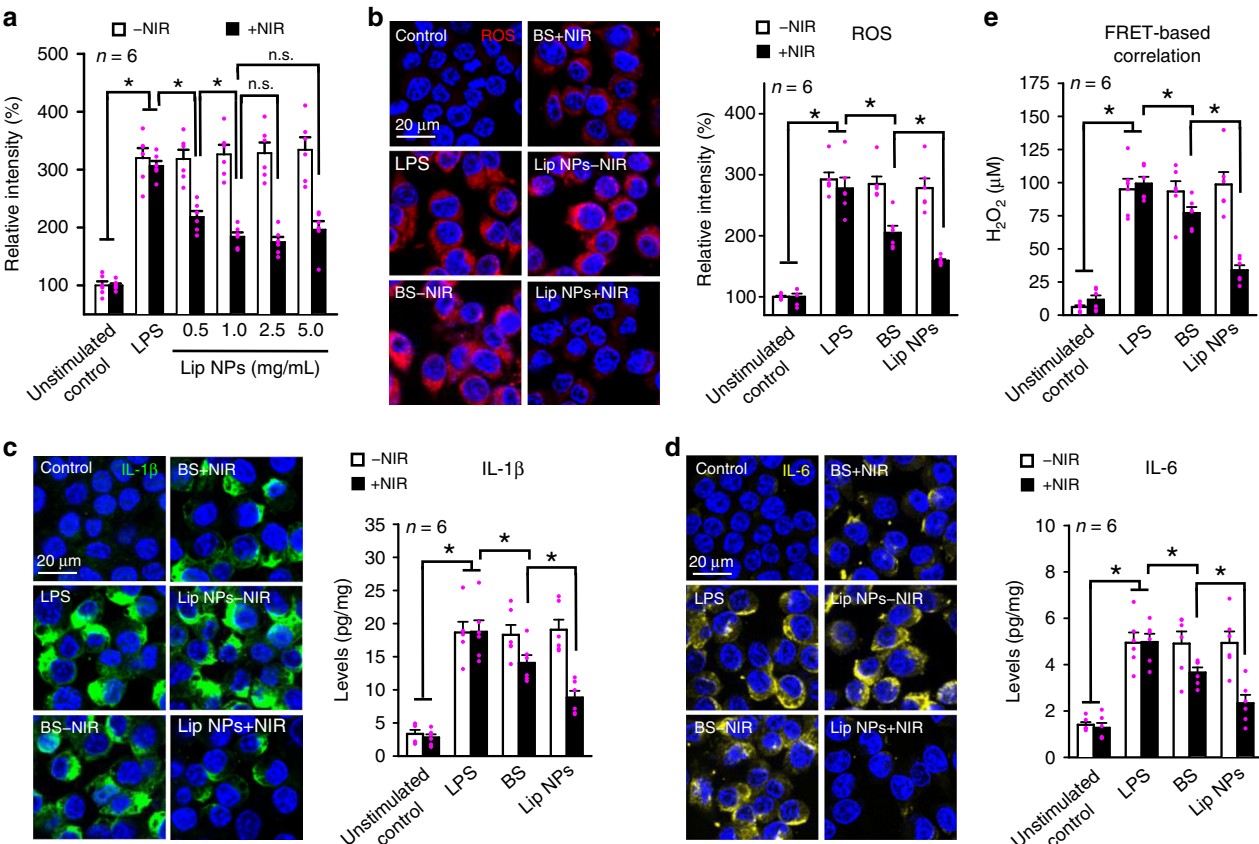

**Fig. 6 In vitro ROS scavenging and anti-inflammation. a** DCF intensities of intracellular ROS levels in LPS-stimulated RAW264.7 cells following treatment with various concentrations of Lip NPs−NIR or Lip NPs + NIR. CLSM images of **b** cellular ROS, **c** IL-1β, **d** IL-6, and their corresponding intensities after various treatments. **e** Concentrations of remaining ROS following various treatments, estimated from linear correlation curve of green UCL ($F/F_0$) intensity against concentration of $H_2O_2$. Data are represented as mean ± SE. Stars indicate significance in the two-tailed unpaired Student's $t$ test; *$P < 0.05$. Each pink dot represents one observed data point. Source data are provided as Source Data file.

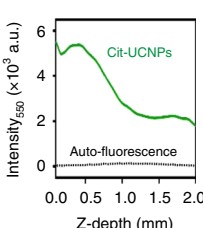 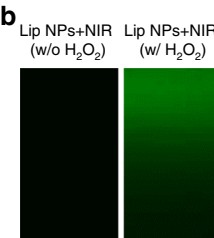 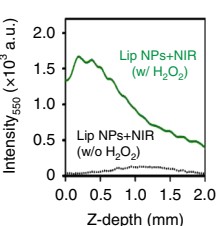 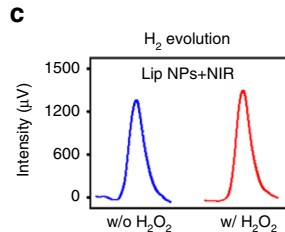

**Fig. 7 Ex vivo ROS detection and H$_2$ generation. a** Fluorescence images and intensities of tissue autofluorescence and NIR light propagation in porcine skin tissue injected with Cit-UCNPs. **b** Fluorescence images and intensities of NIR light propagation in porcine skin tissues injected with Lip NPs in absence/presence of H$_2$O$_2$ and **c** their H$_2$ evolution profiles. Source data are provided as Source Data file.

H$_2$, triggered by the Cit-UCNP-converted red UCL, whose intensity remained relatively constant and was barely affected by the conjugation with AuNPs (Fig. 7c). This observation suggests the capacity of the Lip NPs in the NIR-to-vis-driven photosynthesis of gaseous H$_2$ in situ in both normal and ROS-rich environments. H$_2$ exerts no toxicity even at high doses[37].

In summary, the above results strongly support the claim that the NIR-to-vis-excited Cit-UCNP that is incorporated into the as-proposed Lip NPs can act as a remotely controlled nanotransducer, generating visible UCL in situ, which can be used for determination of the local ROS concentration and concurrent activation of the photosynthesis of gaseous H$_2$, effectively reducing ROS levels in diseased cells. The engineering of such a bioinspired nanoplatform, which integrates diagnosis, therapy, and the monitoring of therapeutic effects in Lip NPs, can greatly help to reestablish ROS homeostasis, likely preventing the development of various human diseases.

## Methods

**Materials**. Oleic acid, cysteamine, Chl*a*, and LPS were obtained from Sigma-Aldrich (St. Louis, MO, USA). NOBF$_4$ and sodium citrate were acquired from Acros (Somerville, NJ, USA). AuNPs were purchased from Nanocomposix (San Diego, CA, USA). The lipids of DPPC and DSPE-PEG2000 were procured from Avanti Polar Lipids (Alabaster, AL, USA). The mouse macrophage cell line (RAW264.7) was obtained from the Bioresource Collection and Research Center, Food Industry Research and Development Institute (Hsinchu, Taiwan). All other chemicals and reagents were of analytical grade.

**Syntheses of OA-UCNPs and Cit-UCNPs**. Ln(CF$_3$COO)$_3$ (Ln = Yb, Er) and Ca (CF$_3$COO)$_2$ were prepared by reacting Ln$_2$O$_3$ (10 mmol) and CaCO$_3$ (10 mmol), respectively, with trifluoroacetic acid (TFA) solution at 110 °C. Then, Yb (CF$_3$COO)$_3$ (0.98 mmol), Er(CF$_3$COO)$_3$ (0.02 mmol), and CF$_3$COONa (1 mmol) were added to a mixture of oleic acid (OA, 10 mmol), oleylamine (10 mmol), and 1-octadecane (ODE, 20 mmol), and reacted at 110 °C (30 min) and 300 °C (30 min) in an inert atmosphere. The resulting NaYbF$_4$:Er core was washed with cyclo-hexane, and then re-dispersed in a mixture of OA (20 mmol) and ODE (20 mmol); Ca(CF$_3$COO)$_2$ (4 mmol) was then added. Subsequently, the solution was heated to 90 °C to remove the cyclohexane and then reacted at 110 °C (30 min) and 300 °C (30 min) in an inert atmosphere. After cooling, an excess of ethanol was added to yield OA-UCNPs.

The Cit-UCNPs were prepared by treating OA-UCNPs with sodium citrate using a ligand exchange method with slight modifications[27]. Briefly, the oleic acid ligands that had been capped on the surfaces of UCNPs were removed by treating them with NOBF$_4$ (0.01 M); the NOBF$_4$-treated UCNPs were then coated with citrate. Ligand exchange from OA-UCNPs to Cit-UCNPs was confirmed by FT−IR (Perkin-Elmer, Buckinghamshire, UK). The TGA (SDT Q600, TA Instruments, New Castle, DE, USA) was used to measure the lost masses of Cit and lipoic acid that capped the surfaces of UCNPs and AuNPs, respectively.

**Synthesis of ROS-responsive TK-based linker**. Cysteamine (26 mmol) together with triethylamine (39 mmol) were dissolved in methanol (25 mL) and then reacted with ethyl trifluoroacetate (31 mmol); the resulting trifluoroacetate (TFA)-protected cysteamine was extracted using ethyl acetate. Next, the extracted TFA-protected cysteamine (6.3 mmol) underwent a Michael addition reaction with 2-methoxypropene (2.5 mmol) in the presence of p-toluenesulfonic acid mono-hydrate (0.8 mmol). The TFA groups of the products were then removed using 6 M NaOH to yield the TK-based linker, which was analyzed by $^1$H NMR spectroscopy (Bruker Avance 500, Frankfurt, Germany).

**Preparation/characterization of nanocomplexes and Lip NPs**. The nanocomplexes were prepared using a typical two-step method[38,39]. First, the as-synthesized Cit-UCNPs (3.6 µmol) and TK-based linker (7.2 µmol) in a molar ratio of 1:2 were dissolved in tetrahydrofuran (THF) and underwent the 1-ethyl-3-(3-dimethyla-minopropyl) carbodiimide (EDC, 7.2 µmol)/*N*-hydroxysuccinimide (NHS, 7.2 µmol) reaction for 48 h at room temperature. After the THF had been removed using a rotavap, the Cit-UCNP-TK was dissolved in dimethylformamide (DMF, 1 mL) and dialyzed against deionized (DI) water for 3 days. The resulting aqueous Cit-UCNP-TK was stored at 4 °C until use. Then, the Cit-UCNP-TK (1.3 µmol) was coupled with AuNPs (1.3 µmol) in a molar ratio of 1:1 in the presence of EDC/NHS (1.3 µmol each). The AuNPs used herein, which were capped with lipoic acid and had a diameter of ca. 5.5 nm, were obtained commercially. The obtained nanocomplexes were then dialyzed against DI water for 3 days.

Lip NPs were prepared by the thin-film hydration technique[40]. Briefly, DPPC, cholesterol, and DSPE-PEG2000 in a molar ratio of 6:4:0.5 were mixed with Chl*a* (60 µM) in chloroform; then, lipid film was obtained by using the rotavap to remove the organic solvent. Hydration with an aqueous solution of nanocomplexes under sonication yielded the Lip NPs. Free nanocomplexes were removed by dialysis against phosphate-buffered saline (PBS) for 3 days.

The morphologies of OA-UCNPs, Cit-UCNPs, nanocomplexes, and Lip NPs were observed using TEM (JEM-2100F, JEOL, Tokyo, Japan). The composition of nanocomplexes was analyzed using STEM that was equipped with EDX for linescans (JEM-ARM200FTH, JEOL, Tokyo, Japan). The zeta potentials of the Cit-UCNPs, AuNPs, Cit-UCNP-TK, nanocomplexes, and Lip NPs were obtained by dynamic light scattering (DLS, Zetasizer 3000HS, Malvern Instruments, Worcestershire, UK).

**Sensitivity of nanocomplexes to H$_2$O$_2$**. To evaluate the sensitivity of nanocom-plexes to H$_2$O$_2$, test nanocomplexes were incubated with H$_2$O$_2$ at various con-centrations (0–50 µM) for 30 min and their fluorescence excitation spectra were recorded using a fluorescence spectrometer (Hitachi F-2500, Tokyo, Japan).

**Evolution of gaseous H$_2$**. The profiles of the gaseous H$_2$ that accumulated during NIR laser exposure were obtained using gas chromatography. The gaseous H$_2$ that was evolved from each test sample under NIR laser irradiation (980 nm, 500 mW/cm$^2$) for various periods (0, 5, 10, 20, 30, or 40 min) was transferred to the air in a test vial. One milliliter of the gas mixture was collected from the test vial using an airtight syringe and subjected to a gas chromatography device (GC-BID, Shimadzu Scientific Instruments, Kyoto, Japan) to determine the concentration of gaseous H$_2$[5].

**Stability of Lip NPs**. Since particle size is one of the indicators of the stability of liposomal formulations[41], the sizes of Lip NPs before and after they had been treated with NIR laser irradiation (980 nm, 500 mW/cm$^2$) and/or H$_2$O$_2$ (50 µM) for 30 min were examined using DLS.

**Cytotoxicity of Lip NPs**. RAW264.7 cells were used to evaluate the cytotoxicity of Lip NPs. The cells were seeded in 96-well plates that contained Dulbecco's mod-ified Eagle's medium, supplemented with 10% fetal bovine serum (HyClone Laboratories, Logan, UT, USA) at $5 \times 10^4$ cells per well. Twenty-four hours later, the cells were incubated with Lip NPs at various concentrations (0−5 mg/mL). Following incubation for another 24 h, the cell viability in the absence/presence of NIR laser irradiation for 30 min was obtained using a WST-1 assay kit (TaKaRa, Otsu, Japan).

**Levels of cellular ROS**. To assess cellular ROS levels, RAW264.7 cells were seeded at $5 \times 10^4$ cells per well in a 96-well dark plate for 24 h. The cells were then stimulated with LPS (1 µg/mL) for 6 h to induce the overproduction of ROS[8]. These LPS-stimulated cells were treated with BS or Lip NPs (1 mg/mL) in the absence/presence of NIR irradiation for 30 min. Then, these treated cells were incubated with 20 µM DCFDA (Abcam, Cambridge, MA, USA) at 37 °C and 5% CO$_2$ for 30 min and then immediately analyzed using a spectrophotometer (SpectraMax M5, Molecular Devices, Sunnyvale, CA, USA)[42]. To visualize the intracellular ROS,

a cell-permeable fluorogenic probe (5 μM; CellROX™, Molecular Probes, Eugene, OR, USA) was used to detect the ROS that were generated in LPS-stimulated RAW264.7 cells ($5 \times 10^4$ cells per well) using CLSM (LSM 780, Carl Zeiss, Jena, Germany)[43]. A commercially available assay kit (ROS-Glo $H_2O_2$ Assay, Promega, Madison, WI, USA) was used to measure extracellular $H_2O_2$ concentrations in cell culture systems following various treatments.

**Levels of cellular proinflammatory cytokines.** The suppression of the over-production of IL-1β and IL-6 in the LPS-stimulated RAW264.7 cells by test samples (BS or Lip NPs in the absence/presence of NIR irradiation for 30 min) was evaluated. At the end of each cell culture experiment, the test cells were lysed in an immunoprecipitation lysis buffer (Thermo Fisher Scientific, Waltham, MA, USA), and the cell lysates were centrifuged at $18,000 \times g$ for 10 min at 4 °C. The super-natants were collected and analyzed using ELISA kits (IL-1β and IL-6 Quantikine ELISA Kits, R&D Systems, Minneapolis, MN, USA).

Immunocytochemical staining was conducted to visualize the expressions of cellular proinflammatory cytokines IL-1β and IL-6. Briefly, the test cells were washed using PBS and then fixed in 4% paraformaldehyde. The fixed cells were blocked using 5% goat serum for 1 h at 37 °C and incubated overnight with monoclonal antibodies for IL-1β (1:200, ab9722, Abcam) and IL-6 (1:150, ab9324, Abcam) in 5% goat serum at 4 °C. Subsequently, the cells were washed again and incubated with suitable secondary antibodies (1:500, A11006 for IL-1β and A11029 for IL-6, Invitrogen, Carlsbad, CA, USA) for 2 h at 37 °C in the dark. Following three washes with PBS, the nuclei were stained with DAPI (1:1000, D1306, Invitrogen). Photomicrographs were obtained by CLSM.

**Determination of NIR penetration depth.** Porcine skin tissues with dimensions of $2\,mm \times 2\,mm \times 2\,mm$ (length × width × height) were used to determine the depth of penetration of the NIR laser (980 nm). Test solutions that contained the Cit-UCNPs or Lip NPs were locally injected from the surface to the deep region of a porcine tissue, which was then placed in a dish with aqueous Cit-UCNPs or Lip NPs and stored overnight[29]. Fluorescence images under irradiation by an external 980 nm NIR laser were obtained using a fluorescence microscope (Zeiss Axio Observer Z1, Gottingen, Germany).

**Statistical analysis.** All results are presented as mean ± SE. The Student's $t$ test was performed to compare the means of experimental groups. Differences were considered to be statistically significant at $P < 0.05$.

**Reporting summary.** Further information on research design is available in the Nature Research Reporting Summary linked to this article.

## Data availability
The source data for main Figs. 3b, d, e, 4a, 5a, b, d, e, 6a–e, 7a–c, Supplementary Fig. 1, and Supplementary Table 1 are provided in a "Source Data" file.

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

## Acknowledgements

The authors would like to thank the Ministry of Science and Technology (MOST 107-2119-M-007-016, 107-3017-F-007-002), the Ministry of Education (MOE 107QR001I5), and the National Health Research Institutes (NHRI-EX107-10522EI) of Taiwan, ROC for financially supporting this research.

## Author contributions

W.-L.W., Y.C., and H.-W.S. conceived and designed the experiments. W.-L.W., B.T., C.K., and D.W. synthesized and characterized the nanocomplexes. M.-Y.L. conducted the STEM experiment and interpreted the data. B.T. and Q.C. carried out the TEM examination/analysis. W.-L.W. and Y.-J.L. performed the $H_2$ evolution experiment and the in vitro cell culture study. W.-L.W., B.T., Y.-J.L., and C.K. prepared figures and analyzed the data. W.-L.W., B.T., Y.-J.L., Y.C., and H.-W.S. wrote the draft manuscript. All authors approved the final version of the manuscript.

## Competing interests

The authors declare no competing interests.
