## [Peer Review File · Nature Communications]

Reviewers' comments:

Reviewer #1 (Remarks to the Author):

In this manuscript, Wan et al. presented a chlorophyll-loaded liposomal formulation encapsulating citrate upconversion nanoparticles for remotely controlled generation of visible light and photosynthesis of H₂ from local ROS. The authors demonstrated a linear relationship between fluorescence signal from the nanoformulation and the concentration of hydrogen peroxide. The formulation detected ROS with high sensitivity, and effectively converted ROS to hydrogen gas. In a porcine skin ex vivo model, the formulation showed deep tissue detection of local ROS, which was desirable for diagnostic applications. Overall, this work is of high interest and the reported platform shows great promise. Below are some minor technical comments for the authors' consideration.

1. Stability of the liposomal formulation should be studied. Would the liposomes be damaged upon ROS scavenging and H₂ generation?
2. Fluorescence image in Figure 2C raises some concerns. 1) The particles shown had different size distribution than particles in Figure 2A-2B. Wide distribution of nanoparticle size often indicates unpredictable performance in biological systems. 2) Chlorophyll should only be embedded in the lipid bilayer. However, the current image showed clusters of green fluorescence outside of the liposomes. It is difficult to claim whether the clusters are smaller liposomes or lipid aggregates. Please improve the current data or consider alternative methods to demonstrate chlorophyll loading in liposomes.
3. Correlations in Figure 3B should be further explored. What's the dynamic range of this linear correlation of fluorescence ratio and H₂O₂ concentration? What happens when H₂O₂ concentration is lower than 5 μ M or higher than 50 μ M. More importantly, to validate the idea of ultrasensitive H₂O₂ detection, what is the relevant H₂O₂ concentration in pathological conditions? Please compare the detection limit with disease-relevant H₂O₂ levels.
4. How to validate the ROS concentration in Figure 4E? Although a linear correlation of fluorescence ratio and H₂O₂ concentration was shown in Figure 3, it did not confirm linear correlation of fluorescence ratio with all ROS in cell culture systems. It would be more convincing to again compare fluorescence reading with actual measurement of intracellular ROS.
5. Cytotoxicity of the formulation should be further studied. Would the hydrogen gas generation cause cytotoxicity? Please investigate cytotoxicity under NIR + liposome + H₂O₂ treatment.
6. Please include the statistical analysis methods for all analyses performed in the figures.

Reviewer #2 (Remarks to the Author):

Review of Wei-Lin Wan, Bo Tian, Yu-Jung Lin, Chiranjeevi Korupalli, Ming-Yen Lu, Qinghua Cui, Dehui Wan, Yen Chang, and Hsing-Wen Sung Bioinspired Photosynthesis of H₂ Gas Using a Chlorophyll-Loaded Liposomal Nanoplatform that Can In Situ Detect Concentration of ROS and Restore Their Homeostasis

Nature Communications, 2019

General recommendation

The importance of reactive oxygen species as both signal molecules and agents of cellular damage is becoming clear in both animal and plant systems, and therefore methods for detection of ROS are important, and methods to lower the concentrations of ROS can offer medical value in the future.

The manuscript describes a combination of liposomal photon-upconverting nanoparticles, a ROS sensitive linker between a fluorescence emitter and FRET acceptor-quencher nanoparticle together with a chlorophyll a and citrate method used to produce hydrogen gas. The reaction of ROS (hydrogen peroxide) with the ROS-sensitive linker destroys the FRET connection, allowing the red up-conversion luminescence to efficiently excite the chlorophylls, which leads to hydrogen

evolution. Experiments with cell cultures suggests that hydrogen evolving from the system may react with some reactive oxygen species produced during artificial inflammation of the mouse cell cultures treated with bacterial lipopolysaccharide.

The manuscript is very well written, easy to follow and interesting for the general audience. The use of the up-conversion and FRET with the ROS-sensitive linker is a smart idea.

I am a plant biologist and my knowledge of macrophages is too limited to critically evaluate the validity of the used cell line. My main concern is related to the suggested mechanism of hydrogen production. The authors suggest that excitation of chlorophyll a causes the formation of an electron-hole pair, and the hole can be filled with an electron from a citrate complex. Such a mechanism is completely new and needs a lot of work to be confirmed (the cited references provide no evidence for such a mechanism). Firstly, as no electron acceptor for the excited chlorophyll was suggested, the reduction of the excited chlorophyll would result in formation of a chlorophyll a anion radical. However, in photosynthesis, excited chlorophyll a molecules act as reductants, not oxidants as suggested in the manuscript (in Photosystem I, an excited chlorophyll a reduces a ground-state chlorophyll a). Secondly, the photosynthetic electron transfer reactions are not intramolecular reactions within a chlorophyll molecule, but reactions between different molecular species held near to each other by a protein matrix.

The manuscript cannot be published before the mechanism of hydrogen evolution in the system is explained, at least by presenting a plausible hypothesis about the mechanism. Perhaps gold functions as a catalyst here?

I am a bit concerned about the tone used in the conclusions. I think that playing around with hydrogen and ROS using medical compounds may become an option for design of new medicines.

Sincerely yours
Esa Tyystjarvi

Reviewer #3 (Remarks to the Author):

This manuscript focuses on developing nanoparticles that can generate hydrogen gas, after near infra red excitation. The nanoparticles are composed of a lipid bilayer that has chlorophyll embedded in it, in addition, the liposomes encapsulate an up-converting lanthanide doped nanoparticle that has been conjugated to gold nanoparticles via an ROS responsive thioketal linkage. Data is presented demonstrating that the nanoparticles can image hydrogen peroxide, can generate hydrogen gas and were not toxic. In addition, cell culture data is presented demonstrating that nanoparticles can scavenge ROS and lower the amount of cytokines generated from LPS stimulation.

Although the concept of using nanoparticles to generate hydrogen gas is novel, the nanoparticles presented here are so complicated that their synthesis will be impossible to reproduce by other laboratories. The manuscript is also challenging to understand because of the complexity of the material. A few steps of the synthesis also did not make sense. How is the thioketal incorporated in the nanoparticles, it appears this occurs by doing sequential NHS couplings with the up-converting particle and then the gold nanoparticle. Usually, these types of reaction schemes generate large amounts of cross-linking and precipitation, and it is surprising that this not observed here. In addition, the thioketal linkage is usually cleaved by radical oxidants, not by hydrogen peroxide, and the studies here appear to use hydrogen peroxide to cleave the thioketal linkage.

Overall, this manuscript, in its current state, lacks the significance needed for publication in Nature Communications. A revised synthetic protocol for the nanoparticles is needed, in which all steps can be characterized and quantified.

In addition, the novel element of this manuscript appears to be the use of near-IR light to generate hydrogen gas. A revised manuscript should focus on this point exclusively, as opposed to also imaging ROS, which can already be done by multiple technologies. Finally, an in vivo animal model demonstrating the ability of these nanoparticles to rescue mice from inflammatory diseases

needs to be performed, and some advantage of using hydrogen gas over other antioxidants also needs to be demonstrated.

Point-to-Point Response Letter

Reviewer #1

“In this manuscript, Wan et al. presented a chlorophyll-loaded liposomal formulation encapsulating citrate upconversion nanoparticles for remotely controlled generation of visible light and photosynthesis of H₂ from local ROS. The authors demonstrated a linear relationship between fluorescence signal from the nanoformulation and the concentration of hydrogen peroxide. The formulation detected ROS with high sensitivity, and effectively converted ROS to hydrogen gas. In a porcine skin ex vivo model, the formulation showed deep tissue detection of local ROS, which was desirable for diagnostic applications. Overall, this work is of high interest and the reported platform shows great promise. Below are some minor technical comments for the authors’ consideration.”

Re: Thank you very much for your encouraging comments.

1. *“Stability of the liposomal formulation should be studied. Would the liposomes be damaged upon ROS scavenging and H₂ generation?”*

Re: Thank you for the constructive suggestion. Since the particle size is one of the indicators of stability of the liposomal formulations [*Cytometry* 2000, 39, 166–171], the sizes of the Lip NPs before and after various treatments have been examined using dynamic light scattering. As shown in Table S1, the sizes of the Lip NPs before and after treatment do not differ significantly ($P > 0.05$), suggesting that the liposomal formulation is stable upon ROS scavenging and H₂ generation. The method that was used in this additional experiment and the results obtained are included in the fourth paragraph on page 19 and the second paragraph on page 11, respectively.

Table S1. Sizes of Lip NPs before and after various treatments (n = 6).

Before Treatment	NIR Treatment	H ₂ O ₂ Treatment	NIR+H ₂ O ₂ Treatment
209.8 ± 12.3 nm	217.0 ± 15.3 nm	193.4 ± 9.3 nm	193.0 ± 16.3 nm

2. *“Fluorescence image in Figure 2C raises some concerns. 1) The particles shown had different size distribution than particles in Figure 2A-2B. Wide distribution of nanoparticle size often indicates unpredictable performance in biological systems. 2) Chlorophyll should only be embedded in the lipid bilayer. However, the current image showed clusters of green fluorescence outside of the liposomes. It is difficult to claim whether the clusters are smaller liposomes or lipid aggregates. Please improve the current data or consider alternative methods to demonstrate chlorophyll loading in liposomes.”*

Re: Owing to limited resolution of the light microscope (CLSM), the objective cannot be less than half of the wavelength of visible light (0.4–0.7 μm) [*Proc. Natl. Acad. Sci. U. S. A.* 2006, 103, 4797–4798]. Therefore, the Lip NPs that had not been through the extrusion process, which included particles with diameters of 200–300 nm and 2–3 μm (please see the particle distribution blow), were used in our CLSM experiment (Figure 2C), while those that had

been through the extrusion process were used for the rest of the experiments, including the samples that were used for the STEM study (Figure 2B). As pointed out by this reviewer, the clusters of green fluorescence were actually smaller liposomes. To improve the quality of the CLSM image, the samples of Lip NPs that have undergone through centrifugation (3,000 rpm for 5 min) are now used; please see the results below. Figure 2C now presents a new CLSM image, and its figure caption has been slightly modified (please see page 9).

3. *“Correlations in Figure 3B should be further explored. What’s the dynamic range of this linear correlation of fluorescence ratio and H₂O₂ concentration? What happens when H₂O₂ concentration is lower than 5 μM or higher than 50 μM. More importantly, to validate the idea of ultrasensitive H₂O₂ detection, what is the relevant H₂O₂ concentration in pathological conditions? Please compare the detection limit with disease-relevant H₂O₂ levels.”*

Re: Local extracellular concentrations of H₂O₂ under normal physiological conditions are in the range of 0.5–7 μM, while those under pathological conditions are elevated as high as 10–50 μM [*J. Am. Chem. Soc.* 2014, 136, 874–877]. This has been clarified in the first paragraph on page 8. According to Figures 3B and 3C, the range of the linear correlation of the fluorescence ratio with H₂O₂ concentration is 10–50 μM H₂O₂, which is in the detection range of disease-relevant H₂O₂ levels. This concern has now been addressed in the second paragraph on page 10.

4. *“How to validate the ROS concentration in Figure 4E? Although a linear correlation of fluorescence ratio and H₂O₂ concentration was shown in Figure 3, it did not confirm linear correlation of fluorescence ratio with all ROS in cell culture systems. It would be more convincing to again compare fluorescence reading with actual measurement of intracellular ROS.”*

Re: ROS in solution is known to be reactive, and so has a short half-life [Nat. Chem. Biol. 2011, 7, 504–511]. In cells, enzymatic and non-enzymatic reactions can convert ROS to H₂O₂, which has a relatively long half-life and can diffuse out of the cells, making H₂O₂ a good marker of oxidative stress [Adv. Mater. 2016, 28, 8755–8759; Anal. Chem. 2010, 82, 2165–2169]. This issue has been addressed in the first paragraph on page 8. In Figure 4E, the levels of ROS [or to be more exact H₂O₂, as pointed out by this reviewer, so the y-axis of this figure has been changed to ROS (H₂O₂)] that remained in the cell culture media following various treatments, were estimated from the linear correlation of the fluorescence ratio with the concentration of H₂O₂, which was shown in Figure 3C. To verify the H₂O₂ concentration in cell culture systems, an additional experiment using a commercially available assay kit (ROS-Glo H₂O₂ Assay, Promega, Madison, WI, USA) was carried out to measure the actual extracellular H₂O₂ concentrations, as suggested. As displayed in Figure S1, the trends obtained using these two different methods were similar, suggesting that the FRET-based correlation method that was proposed herein may be used to estimate the H₂O₂ levels in cell culture systems. This fact has been clarified in the second paragraph on page 20 (Method) and the last paragraph on page 13 (Results and Discussion).

Concentrations of ROS (H₂O₂) that remained following various treatments, estimated by FRET-base correlation method (left, Figure 4E) and commercially available assay (right, Figure S1).

5. “Cytotoxicity of the formulation should be further studied. Would the hydrogen gas generation cause cytotoxicity? Please investigate cytotoxicity under NIR + liposome + H₂O₂ treatment.”

Re: H₂ is an inert gas at body temperature. It exhibits no cytotoxicity even at high concentrations [Med. Gas Res. 2016, 6, 219–222]. Please see the last paragraph on page 15. As suggested, an additional experiment is performed to investigate the cytotoxicity under NIR + liposome (Lip NPs) + H₂O₂ (generated by LPS-treated cells) treatment. As revealed by the results below, no significant cytotoxicity was detected under NIR + liposome + H₂O₂ treatment relative to the untreated control ($P > 0.05$).

6. “Please include the statistical analysis methods for all analyses performed in the figures.”

Re: As suggested, the method of statistical analysis that was used in each study has been provided. Please refer to the captions in Figures 3 and 4.

Reviewer #2

“The importance of reactive oxygen species as both signal molecules and agents of cellular damage is becoming clear in both animal and plant systems, and therefore methods for detection of ROS are important, and methods to lower the concentrations of ROS can offer medical value in the future. The manuscript describes a combination of liposomal photon-upconverting nanoparticles, a ROS sensitive linker between a fluorescence emitter and FRET acceptor-quencher nanoparticle together with a chlorophyll a and citrate method used to produce hydrogen gas. The reaction of ROS (hydrogen peroxide) with the ROS-sensitive linker destroys the FRET connection, allowing the red up-conversion luminescence to efficiently excite the chlorophylls, which leads to hydrogen evolution. Experiments with cell cultures suggests that hydrogen evolving from the system may react with some reactive oxygen species produced during artificial inflammation of the mouse cell cultures treated with bacterial lipopolysaccharide. The manuscript is very well written, easy to follow and interesting for the general audience. The use of the up-conversion and FRET with the ROS-sensitive linker is a smart idea.”

Re: Thank you very much for your encouraging comments.

1. “I am a plant biologist and my knowledge of macrophages is too limited to critically evaluate the validity of the used cell line. My main concern is related to the suggested mechanism of hydrogen production. The authors suggest that excitation of chlorophyll a causes the formation of an electron-hole pair, and the hole can be filled with an electron from a citrate complex. Such a mechanism is completely new and needs a lot of work to be confirmed (the cited references provide no evidence for such a mechanism).”

Re: Photocatalytic hydrogen production, which typically involves a photosensitizer (chlorophyll a, Chl_a, in our case), a proton-reducing catalyst (gold nanoparticles, AuNPs),

and a sacrificial electron donor (citrate), has been widely used in artificial photosynthetic systems for the efficient utilization of solar energy, solving energy-related and environmental problems [*ACS Appl. Energy Mater.* 2018, 1, 2813–2820; *Nat. Commun.* 2018, 9, 4009; *J. Am. Chem. Soc.* 2018, 140, 1423–1427]. The mechanism of hydrogen production in our artificial photosynthetic system (Lip NPs) has now been described clearly, and more relevant references have been cited. Please see the last paragraph on page 5.

2. *Firstly, as no electron acceptor for the excited chlorophyll was suggested, the reduction of the excited chlorophyll would result in formation of a chlorophyll a anion radical. However, in photosynthesis, excited chlorophyll a molecules act as reductants, not oxidants as suggested in the manuscript (in Photosystem I, an excited chlorophyll a reduces a ground-state chlorophyll a)”*

Re: We apologize for the confusion. In our artificial photosynthetic system, when red UCL at 660 nm is harvested by the photosensitizer (Chla), a photo-excited electron is generated, which is then transferred to the proton-reducing catalyst (AuNPs, which also serve as an electron acceptor). Chla in the excited state can be reduced by the oxidation of a sacrificial electron donor (citrate) to accomplish the cycle of photocatalytic hydrogen production [*Environ Technol.* 2016, 37, 2687–2693; *Energy Fuels* 2003, 17, 1641–1644]. Scheme 2 has now been modified slightly to illustrate better the mechanism of hydrogen production by Lip NPs.

3. *“Secondly, the photosynthetic electron transfer reactions are not intramolecular reactions within a chlorophyll molecule, but reactions between different molecular species held near to each other by a protein matrix.”*

Re: We apologize for the confusion. In fact, in our artificial photosynthetic system, we used Chla, rather than the whole photosystem that is used in the natural photosynthetic system, as a photosensitizer. As soon as the photo-excited electron is generated, it is rapidly transferred to the adjacent proton-reducing catalyst (AuNPs). Therefore, the electron transfer in our system does not require a protein matrix.

4. *“The manuscript cannot be published before the mechanism of hydrogen evolution in the system is explained, at least by presenting a plausible hypothesis about the mechanism. Perhaps gold functions as a catalyst here?”*

Re: The mechanism of hydrogen evolution in our artificial photosynthetic system (Lip NPs) has been further elucidated as described above. As suggested, an additional experiment to compare the amounts of gaseous hydrogen that are evolved from the Lip NPs with and without AuNPs has been performed. As shown in the figure below, at 30 min following exposure to the NIR laser, a significant amount of gaseous hydrogen was observed by the Lip

NPs with AuNPs, while no hydrogen evolution was observed from the Lip NPs without AuNPs, supporting the claim that AuNPs function as a proton-reducing catalyst in the artificial photosynthetic system.

Reviewer #3

“This manuscript focuses on developing nanoparticles that can generate hydrogen gas, after near infrared excitation. The nanoparticles are composed of a lipid bilayer that has chlorophyll embedded in it, in addition, the liposomes encapsulate an up-converting lanthanide doped nanoparticle that has been conjugated to gold nanoparticles via an ROS responsive thioketal linkage. Data is presented demonstrating that the nanoparticles can image hydrogen peroxide, can generate hydrogen gas and were not toxic. In addition, cell culture data is presented demonstrating that nanoparticles can scavenge ROS and lower the amount of cytokines generated from LPS stimulation. Although the concept of using nanoparticles to generate hydrogen gas is novel, the nanoparticles presented here are so complicated that their synthesis will be impossible to reproduce by other laboratories. The manuscript is also challenging to understand because of the complexity of the material.”

- 1. “A few steps of the synthesis also did not make sense. How is the thioketal incorporated in the nanoparticles, it appears this occurs by doing sequential NHS couplings with the up-converting particle and then the gold nanoparticle. Usually, these types of reaction schemes generate large amounts of cross-linking and precipitation, and it is surprising that this not observed here.”*

Re: We agree with this reviewer that most reaction schemes for the preparation of nanocomplexes may cause significant precipitation; however, this precipitation can be minimized by controlling the reaction conditions. In the present study, nanocomplexes were prepared using a typical two-step method that has been previously reported in the literature [*J. Mater. Chem. B* 2016, 4, 4675–4682; *Eur. Polym. J.* 2016, 85, 38–52]. In our preliminary study, the molar ratio of Cit-UCNP:thioketal (TK)-based linker (1:2, 1:3, 1:5, or 1:10) in the first-step synthesis was optimized. No significant precipitation occurred when Cit-UCNPs and the TK-based linker were in a molar ratio of 1:2 (please see the results below); this formulation was thus used in the subsequent study. In the second-step synthesis of nanocomplexes, AuNPs were conjugated only to Cit-UCNP-TK, and could not form AuNP-TK-AuNP complexes, owing to the absence of free TK linkers; therefore, no precipitation occurred (see the result below).

(A) Photograph of solutions of Cit-UCNP-TK that were prepared at various molar ratios of Cit-UCNP:TK-based linker that were obtained in the first-step synthesis. (B) Photograph of solution of Cit-UCNP-TK-AuNPs that was obtained in the second-step synthesis.

2. *“In addition, the thioketal linkage is usually cleaved by radical oxidants, not by hydrogen peroxide, and the studies here appear to use hydrogen peroxide to cleave the thioketal linkage.”*

Re: We agree with this reviewer that thioketal moieties can be cleaved by oxidative means [Adv. Sci. 2017, 4, 1600124]. However, thioketals can also be oxidized in the presence of H₂O₂, producing two thiol groups and one ketone [Adv. Biosys. 2017, 1, 1700084]. H₂O₂ has been used to cleave thioketal linkages in the literature [J. Am. Chem. Soc. 2018, 140, 24, 7373-7376; Eur. Polym. J. 2016, 85, 38–52; Nano Res. 2019, 12, 5, 999–1008; Nanomedicine 2015, 10, 17, 2709–2723].

3. *“A revised synthetic protocol for the nanoparticles is needed, in which all steps can be characterized and quantified.”*

Re: Thank you for the constructive suggestions. The protocols for the synthesis of ROS-responsive thioketal-based linker and nanocomplexes have now been described in detail. Please refer to the last paragraph on page 17 and the first and second paragraphs on page 18.

4. *“In addition, the novel element of this manuscript appears to be the use of near-IR light to generate hydrogen gas. A revised manuscript should focus on this point exclusively, as opposed to also imaging ROS, which can already be done by multiple technologies.”*

Re: We apologize for the confusion. The as-proposed Lip NPs can act as a remotely controlled nanotransducer, generating visible UCL (green and red) for simultaneous imaging and therapy *in situ*, upon excitation by NIR light. The generated green UCL is used to detect the local ROS concentration for FRET imaging, and the red UCL induces the photosynthesis of H₂ gas, which scavenges excess ROS, modulating ROS homeostasis. (The same system can perform these two functions concurrently; please see the mechanism illustrated below.) In the study, this FRET imaging technique was demonstrated to be a powerful technique for detecting excess ROS *in vitro* (Figures 3B and 4E) and *ex vivo* (Figure 5). The engineering of such a bioinspired nanoplatform, which integrates diagnosis, therapy, and the monitoring of therapeutic effects in Lip NPs, can greatly help to reestablish ROS homeostasis. Please see the Conclusion (the last paragraph on page 15).

5. “Finally, an *in vivo* animal model demonstrating the ability of these nanoparticles to rescue mice from inflammatory diseases needs to be performed, and some advantage of using hydrogen gas over other antioxidants also needs to be demonstrated.”

Re: In the study, we designed an *ex vivo* model demonstrating the proof-of-principle application of the as-proposed artificial photosynthetic system (Lip NPs). An *in vivo* animal model application is not within the focus of this study. The advantages of using hydrogen gas over other antioxidants have been elaborated further in the second paragraph on page 3.

REVIEWERS' COMMENTS:

Reviewer #1 (Remarks to the Author):

The authors have done an excellent job in revising the manuscript. All of my previous comments have been fully addressed.

Reviewer #2 (Remarks to the Author):

General judgment

The authors have successfully answered to my comments about the mechanism of H₂ evolution but I still have a concern (#1) that must be taken into account. Taking also the minor comment (#2) into account would improve the manuscript.

1) Line 80 and Scheme 2. A ground-state orbital of an excited chlorophyll a (Chl^a*) cannot be filled by an electron from citrate. A reductant of Chl* should be highly reducing (I have ever heard even about existence of such a reduction reaction) while the redox potential of the citrate/citrate⁺ pair is only -180 mV. The production of Chl⁺ necessarily occurs before reduction by citrate. There is no such thing as return to the ground state by filling an empty orbital of an excited chlorophyll a molecule. The result of such a reaction would be a chlorophyll anion, and the in vitro redox potential of the Chl^a/Chl⁻ is around -900 mV. The redox potential of Chl^a*/Chl^a*(-) has not been measured but in the photosynthetic reaction centers, the Chl^a+ /Chl* has a more negative potential than Chl^a+ /Chl^a. The scheme should be citrate + Chl^a (light) -> citrate + Chl^a* -> citrate + Chl^a+ + e -> citrate⁺ + Chl^a + e.

Minor comment:

2) Lines 129-132. From the introductory text in the beginning of the manuscript, the authors obviously know that "ROS" is a collective term that in their case is used to refer to several chemical species. However, the text here sounds as if "ROS" was a single molecular species. The term "ROS (H₂O₂)" on lines 229-234 should also be changed simply to H₂O₂ (and can be changed to "ROS" on lines 238-239, as this is a conclusion that the authors wish to extend to estimation of other ROS from measurements of H₂O₂).

Esa Tyystjärvi

Reviewer #3 (Remarks to the Author):

The authors have addressed the comments of this Reviewer. No additional work or explanations are needed.

Point-to-Point Response Letter

Reviewer #1

“The authors have done an excellent job in revising the manuscript. All of my previous comments have been fully addressed.”

Re: We thank this reviewer for the positive comments on our revised manuscript.

Reviewer #2

“General judgment: The authors have successfully answered to my comments about the mechanism of H₂ evolution but I still have a concern (#1) that must be taken into account. Taking also the minor comment (#2) into account would improve the manuscript.”

“1) Line 80 and Scheme 2. A ground-state orbital of an excited chlorophyll a (Chla) cannot be filled by an electron from citrate. A reductant of Chl* should be highly reducing (I have ever heard even about existence of such a reduction reaction) while the redox potential of the citrate/citrate+ pair is only -180 mV. The production of Chl+ necessarily occurs before reduction by citrate. There is no such thing as return to the ground state by filling an empty orbital of an excited chlorophyll a molecule. The result of such a reaction would be a chlorophyll anion, and the in vitro redox potential of the Chla/Chl- is around -900 mV. The redox potential of Chla*/Chla*(-) has not been measured but in the photosynthetic reaction centers, the Chla+/Chl* has a more negative potential than Chla+/Chla. The scheme should be citrate + Chla (light) -> citrate + Chla* -> citrate + Chla+ + e -> citrate+ + Chla + e.”*

Re: We thank this reviewer for the comments; in response to which, we have modified the second paragraph on page 5 and Scheme 2 (now Figure 2) as shown below.

Figure 2 presents a potential mechanism of the photocatalytic evolution of hydrogen. When red UCL at 660 nm is harvested by Chla (a photosensitizer), the latter becomes excited (Chla*). The photo-excited electrons that are released from the Chla* are rapidly transferred to the AuNPs (an electron acceptor and a proton-reducing catalyst) that are conjugated with the Cit-UCNP. The AuNPs then collect protons from citrate (a sacrificial electron donor), which caps the UCNP, and hydrogen is thus evolved, locally scavenging the excess ROS. The oxidized Chla (after the loss of an electron, Chla⁺) can be reduced by its acceptance of an electron from citrate, returning to its ground state.

Minor comment:

2) Lines 129-132. From the introductory text in the beginning of the manuscript, the authors obviously know that “ROS” is a collective term that in their case is used to refer to several chemical species. However, the text here sounds as if “ROS” was a single molecular species. The term “ROS (H_2O_2)” on lines 229-234 should also be changed simply to H_2O_2 (and can be changed to “ROS” on lines 238-239, as this is a conclusion that the authors wish to extend to estimation of other ROS from measurements of H_2O_2 .”

Re: We thank this reviewer for the suggestions. We have now changed the term ROS (H_2O_2) to H_2O_2 on lines 229–234, and ROS (H_2O_2) to ROS on lines 238–239 as suggested.

Reviewer #3

“The authors have addressed the comments of this Reviewer. No additional work or explanations are needed.”

Re: We thank this reviewer for the positive comments on our revised manuscript.